# CoT: Cooperative Training for Generative Modeling of Discrete Data

## Abstract

We propose Cooperative Training (CoT) for training generative models that measure a tractable density for discrete data. CoT coordinately trains a generator $G$ and an auxiliary predictive mediator $M$. The training target of $M$ is to estimate a mixture density of the learned distribution $G$ and the target distribution $P$, and that of $G$ is to minimize the Jensen-Shannon divergence estimated through $M$. CoT achieves independent success without the necessity of pre-training via Maximum Likelihood Estimation or involving high-variance algorithms like REINFORCE. This low-variance algorithm is theoretically proved to be superior for both sample generation and likelihood prediction. We also theoretically and empirically show the superiority of CoT over most previous algorithms in terms of generative quality and diversity, predictive generalization ability and computational cost.

## 1 Introduction

Generative modeling is essential in many scenarios, including continuous data modeling (*e.g.* image generation (Goodfellow et al., 2014; Arjovsky et al., 2017), stylization (Ulyanov et al., 2016), semi-supervised classification (Radford et al., 2015)) and sequential discrete data modeling (*e.g.* neural text generation (Bahdanau et al., 2014; Yu et al., 2017; Lu et al., 2018)).

For discrete data with tractable density like natural language, generative models are predominantly optimized through Maximum Likelihood Estimation (MLE), inevitably introducing *exposure bias* (Ranzato et al., 2015), which results in that given a **finite** set of observations, the optimal parameters of the model trained via MLE do not correspond to the ones maximizing the generative quality. Specifically, the model is trained on the data distribution of inputs and tested on a different distribution of inputs, namely, the learned distribution. This discrepancy implies that in the training stage, the model is never exposed to its own errors and thus in the test stage, the errors made along the way will quickly accumulate.

On the other hand, for general generative modeling tasks, an effective framework, named Generative Adversarial Network (GAN) (Goodfellow et al., 2014), was proposed to train an implicit density model for continuous data. GAN introduces a discriminator $D_\phi$ parametrized by $\phi$ to distinguish the generated samples from the real ones. As is proved in (Goodfellow et al., 2014), GAN essentially optimizes an approximately estimated Jensen-Shannon divergence (JSD) between the currently learned distribution and the target distribution. GAN shows promising results in many unsupervised and semi-supervised learning tasks. The success of GAN results in the naissance of a new paradigm of deep generative models, *i.e.* adversarial networks.

However, since the gradient computation requires backpropagation through the generator's output, GAN can only model the distribution of continuous variables, making it non-applicable for generating discrete sequences like natural language. Researchers then proposed Sequence Generative Adversarial Network (SeqGAN) (Yu et al., 2017), which uses model-free policy gradient algorithm to optimize the original GAN objective. With SeqGAN, the expected JSD between current and target discrete data distribution is minimized if the training is perfect. SeqGAN shows observable improvements in many tasks. Since then, many variants of SeqGAN have been proposed to improve its performance. Nonetheless, SeqGAN is not an ideal algorithm for this problem, and current algorithms based on it cannot show stable, reliable and observable improvements that covers all scenarios, according to a previous survey (Lu et al., 2018). The detailed reason will be discussed in detail in Section 2.

In this paper, we propose Cooperative Training (CoT), a novel, low-variance, bias-free algorithm for training likelihood-based generative models on discrete data by directly optimizing a well-estimated Jensen-Shannon divergence. CoT coordinately trains a generative module $G$, and an auxiliary predictive module $M$, called *mediator*, for guiding $G$ in a cooperative fashion. For theoretical soundness, we derive the proposed algorithm directly from the definition of JSD. We further empirically and theoretically demonstrate the superiority of our algorithm over many strong baselines in terms of generative performance, generalization ability and computational performance in both synthetic and real-world scenarios.

## 2 BACKGROUND

**Notations.** $P$ denotes the target data distribution. $\theta$ denotes the parameters of the generative module $G$. $\phi$ denotes the parameters of the auxiliary predictive mediator module $M$. Any symbol with subscript $_g$ and $_m$ stands for that of the generator and mediator, respectively. $s$ stands for a complete sample from the training dataset or a generated complete sequence, depending on the specific context. $s_t$ means the $t$-length prefix of the original sequence, *i.e.* an incomplete sequence of length $t$. $x$ denotes a token, and $x_t$ stands for a token that appears in the $t$-th place of a sequence. Thus $s_t = [x_0, x_1, x_2, \ldots, x_{t-1}]$ while the initial case $s_0$ is $\emptyset$.

### 2.1 MAXIMUM LIKELIHOOD ESTIMATION

Maximum likelihood estimation is equivalent to minimizing the KL divergence using the samples from the real distribution:
$$\min_\theta \mathbb{E}_{s \sim p_{\text{data}}} \left[ -\log G_\theta(s) \right], \tag{1}$$
where $G_\theta(s)$ is the estimated probability of $s$ by $G_\theta$ and $p_{\text{data}}$ is the underlying real distribution.

**Limitations of MLE.** MLE is essentially equivalent to optimizing a directed Kullback–Leibler (KL) divergence between the target distribution $P$ and the currently learned distribution $G$, denoted as $KL(P\|G)$. However, since KL divergence is asymmetric, given *finite* observations this target is actually not ideal. As stated in (Arjovsky & Bottou, 2017), MLE tries to minimize
$$KL(P\|G) = \sum_s P(s) \log \frac{P(s)}{G(s)}. \tag{2}$$

- When $P(s) > 0$ and $G(s) \to 0$, the KL divergence grows to infinity, which means MLE assigns an extremely high cost to the "mode dropping" scenarios, where the generator fails to cover some parts of the data.
- When $G(s) > 0$ and $P(s) \to 0$, the KL divergence shrinks to 0, which means MLE assigns an extremely low cost to the scenarios, where the model generates some samples that do not locate on the data distribution.

Likewise, optimizing $KL(G\|P)$ will lead to exactly the reversed problems of the two situations. An ideal solution is to optimize a **symmetrized** and **smoothed** version of KL divergence, *i.e.* the Jensen-Shannon divergence (JSD), which is defined as
$$JSD(P\|G) = \frac{1}{2}\big(KL(P\|M) + KL(G\|M)\big), \tag{3}$$
where $M = \frac{1}{2}(P + G)$. However, directly optimizing JSD is conventionally considered as an intractable problem. JSD cannot be directly evaluated and optimized since the equally interpolated distribution $M$ is usually considered to be unconstructable, as we only have access to the learned model $G$ instead of $P$.

### 2.2 SEQUENCE GENERATIVE ADVERSARIAL NETWORK

SeqGAN incorporates two modules, *i.e.* the generator and discriminator, parametrized by $\theta$ and $\phi$ respectively, as in the settings of GAN. By alternatively training these two modules, SeqGAN optimizes such an adversarial target:
$$\min_\theta \max_\phi \mathbb{E}_{s \sim p_{\text{data}}} \left[ \log(D_\phi(s)) \right] + \mathbb{E}_{s \sim G_\theta} \left[ \log(1 - D_\phi(s)) \right]. \tag{4}$$

---

**Algorithm 1** Cooperative Training

---

**Require:** Generator $G_\theta$; mediator $M_\phi$; samples from real data distribution $P$; hyper-parameter $N_m$.
1: Initialize $G_\theta$, $M_\phi$ with random weights $\theta, \phi$.
2: **repeat**
3:     **for** $N_m$ steps **do**
4:         Collect two equal-sized mini-batch of samples $\{s_g\}$ and $\{s_p\}$ from $G_\theta$ and $P$, respectively
5:         Mix $\{s_g\}$ and $\{s_p\}$ as $\{s\}$
6:         Update mediator $M_\phi$ with $\{s\}$ via Eq. (9)
7:     **end for**
8:     Generate a mini-batch of sequences $\{s\} \sim G_\theta$
9:     Update generator $G_\theta$ with $\{s\}$ via Eq. (13)
10: **until** CoT converges

---

The objectives of generator $G_\theta$ and discriminator $D_\phi$ in SeqGAN can be formulated as

$$\text{Generator:} \quad \min_\theta -\mathbb{E}_{s \sim G_\theta} \Big[ \sum_{t=1}^n Q_t(s_t, x_t) \cdot \log G_\theta(x_t|s_t) \Big] \tag{5}$$

$$\text{Discriminator:} \quad \max_\phi \mathbb{E}_{s \sim p_{\text{data}}} \left[ \log(D_\phi(s)) \right] + \mathbb{E}_{s \sim G_\theta} \left[ \log(1 - D_\phi(s)) \right], \tag{6}$$

where $s \sim G_\theta = [x_1, ..., x_n]$ denotes a complete sequence sampled from the generator and the action value $Q_t(s_t, x_t) = \mathbb{E}_{s \sim G_\theta(\cdot|s_{t+1})}[D_\phi(s)]$ is the expectation of the discriminator's evaluation on the completed sequences sampled from the prefix $s_{t+1} = [s_t, x_t]$, which can be approximated via Monte Carlo search.

**Limitations of SeqGAN & its Variants.** First, SeqGAN is an algorithm of high variance, which relies on pre-training via Maximum Likelihood Estimation as a variance reduction procedure. Besides, during the adversarial epochs, even if with variance reduction techniques such as Actor-Critic methods (Sutton, 1984), the fact that SeqGAN is essentially based on model-free reinforcement learning makes it a non-trivial problem for SeqGAN to converge well. As a result, SeqGAN usually gets stuck in some fake local optimals. Specifically, although the discriminator can distinguish the samples from the generator easily, it is not able to effectively guide the generator because of the vanishing gradient, as is discussed in a recent survey (Lu et al., 2018). Although this problem can be alleviated by reshaping the reward signals based on the relative rankings of the outputs in a mini-batch (Lin et al., 2017; Guo et al., 2017), they are more technical workarounds than essential solutions.

Second, SeqGAN trained via REINFORCE (Williams, 1992) suffers from the "mode collapse" problem, which is similar to the original GAN. That is to say, the learned distribution "collapses" to the other side of KL divergence, *i.e.* $KL(G\|P)$, which leads to the loss of diversity of generated samples. In other words, SeqGAN trains the model for better generative quality at the cost of diversity.

## 3   Cooperative Training

### 3.1   Motivation

To be consistent with the goal that the target distribution should be well-estimated in both **quality** and **diversity** senses, an ideal algorithm for such models should be able to optimize a symmetric divergence or distance.

For sequential discrete data modeling, since the data distribution is decomposed into a sequential product of finite-dimension multinomial distributions (always based on the softmax form), the failures of effectively optimizing JSD when the generated and real data distributions are distant, as discussed in (Arjovsky et al., 2017), will not appear. As such, to optimize JSD is feasible. However, to our knowledge, no previous algorithms provide a direct, low-variance optimization of JSD. In this paper, we propose Cooperative Training (CoT), as shown in Algorithm 1, to directly optimize a well-estimated unbiased JSD for training such models.

### 3.2 Algorithm Derivation

Each iteration of Cooperative Training mainly consists of two parts. The first part is to train a *mediator* $M_\phi$, which is a density function that estimates a mixture distribution of the learned generative distribution $G_\theta$ and target latent distribution $P = p_{\text{data}}$ as

$$M_\phi \simeq \frac{1}{2}(P + G_\theta). \tag{7}$$

Since the mediator is only used as a density **prediction** module during training, the directed KL divergence is now free from so-called exposure bias for optimization of $M_\phi$. Denote $\frac{1}{2}(P + G_\theta)$ as $M^*$, we have:

**Lemma 1 (Mixture Density Decomposition)**

$$
\begin{aligned}
\nabla_\phi J_m(\phi) &= \nabla_\phi KL(M^* \| M_\phi) \\
&= \nabla_\phi \mathop{\mathbb{E}}_{s \sim M^*} \left[ \log \frac{M^*(s)}{M_\phi(s)} \right] \\
&= \nabla_\phi \left( - \mathop{\mathbb{E}}_{s \sim M^*} [\log M_\phi(s)] \right) \\
&= \nabla_\phi \frac{1}{2} \left( \mathop{\mathbb{E}}_{s \sim G_\theta} [-\log(M_\phi(s))] + \mathop{\mathbb{E}}_{s \sim P} [-\log(M_\phi(s))] \right)
\end{aligned}
\tag{8}
$$

By Lemma 1, for each step, we can simply mix balanced samples from training data and the generator, then train the mediator via Maximum Likelihood Estimation with the mixed samples. The objective $J_m(\phi)$ for the mediator $M$ parametrized by $\phi$ therefore becomes

$$J_m(\phi) = \frac{1}{2} \left( \mathop{\mathbb{E}}_{s \sim G_\theta} [-\log(M_\phi(s))] + \mathop{\mathbb{E}}_{s \sim P} [-\log(M_\phi(s))] \right). \tag{9}$$

Since the objective of MLE is bias-free for predictive purposes, the estimated $M_\phi$ is also bias-free when adopted for estimating JSD. The training techniques and details will be discussed in Section 4.

After each iteration, the mediator is exploited to optimize an estimated Jensen-Shannon divergence for $G_\theta$:

$$
\begin{aligned}
\nabla_\theta J_g(\theta) &= \nabla_\theta \left( - J\hat{S}D(G_\theta \| P) \right) = \nabla_\theta \left( - \frac{1}{2} \left[ KL(G_\theta \| M_\phi) + KL(P \| M_\phi) \right] \right) \\
&= \nabla_\theta \left( - \frac{1}{2} \mathop{\mathbb{E}}_{s \sim G_\theta} \left[ \log \frac{G_\theta(s)}{M_\phi(s)} \right] - \frac{1}{2} \mathop{\mathbb{E}}_{s \sim P} \left[ \log \frac{P(s)}{M_\phi(s)} \right] \right) = \nabla_\theta \left( - \frac{1}{2} \mathop{\mathbb{E}}_{s \sim G_\theta} \left[ \log \frac{G_\theta(s)}{M_\phi(s)} \right] \right).
\end{aligned}
\tag{10}
$$

Note that the gradient Eq. (10) should be performed for only one step because once $G_\theta$ is updated the current mediator's estimation $M_\phi$ becomes inaccurate.

For any sequence or prefix of length $t$, we have:

**Lemma 2 (Markov Backward Reduction)**

$$\nabla_\theta \left( - \frac{1}{2} \mathop{\mathbb{E}}_{s_t \sim G_\theta} \left[ \log \frac{G_\theta(s_t)}{M_\phi(s_t)} \right] \right) \tag{11}$$

$$= \nabla_\theta \left( - \frac{1}{2} \mathop{\mathbb{E}}_{s_{t-1} \sim G_\theta} \left[ \sum_{s_t} G_\theta(s_t | s_{t-1}) \log \frac{G_\theta(s_t | s_{t-1})}{M_\phi(s_t | s_{t-1})} \right] - \frac{1}{2} \mathop{\mathbb{E}}_{s_{t-1} \sim G_\theta} \left[ \log \frac{G_\theta(s_{t-1})}{M_\phi(s_{t-1})} \right] \right). \tag{12}$$

The detailed derivations can be found in the supplementary material. Note that Lemma 2 can be applied recursively. That is to say, given any sequence $s_t$ of arbitrary length $t$, optimizing $s_t$'s contribution to the expected JSD can be decomposed into optimizing the first term of Eq. (12) and solving an isomorphic problem for $s_{t-1}$, which is the longest proper prefix of $s_t$. When $t = 1$, since in Markov decision process the probability for initial state $s_0$ is always 1.0, it is trivial to prove that the final second term becomes 0.

Therefore, Eq. (10) can be reduced through recursively applying Lemma 2. After removing the constant multipliers and denoting the predicted probability distribution over the action space, *i.e.* $G_\theta(\cdot|s_t)$ and $M_\phi(\cdot|s_t)$, as $\pi_g(s_t)$ and $\pi_m(s_t)$ respectively, the gradient $\nabla_\theta J_g(\theta)$ for training generator via Cooperative Training can be formulated as

$$\nabla_\theta J_g(\theta) = \nabla_\theta \mathop{\mathbb{E}}_{s \sim G_\theta} \Big[ \sum_{t=0}^{n-1} \pi_g(s_t)^\top (\log \pi_m(s_t) - \log \pi_g(s_t)) \Big]. \tag{13}$$

For tractable density models with finite discrete action space in each step, the practical effectiveness of this gradient is well guaranteed for the following reasons. First, with a random initialization of the model, the supports of distributions $G_\theta$ and $P$ are hardly disjoint. Second, the first term of Eq. (13) is to minimize the cross entropy between $G$ and $M^*$, which tries to enlarge the overlap of two distributions. Third, since the second term of Eq. (13) is equivalent to maximizing the entropy of $G$, it encourages the support of $G$ to cover the whole action space, which avoids the case of disjoint supports between $G$ and $P$.

The overall objective of CoT can be formulated as finding the maximal entropy solution of

$$\max_\theta \max_\phi \mathop{\mathbb{E}}_{s \sim p_{\text{data}}} [\log(M_\phi(s))] + \mathop{\mathbb{E}}_{s \sim G_\theta} [\log(M_\phi(s))]. \tag{14}$$

Note the strong connections and differences between the optimization objective of CoT (14) and that of GAN (4). Figure 1 illustrates the whole Cooperative Training process.

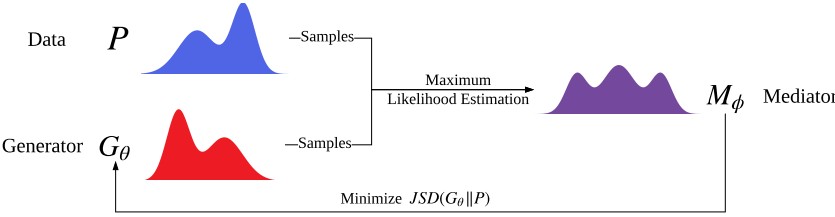

Figure 1: Process of Cooperative Training.

### 3.3 CONVERGENCE ANALYSIS

CoT has theoretical guarantee on its convergence.

**Theorem 3 (Jensen-Shannon Consistency)** *If in each step, the mediator $M_\phi$ of CoT is trained to be optimal, i.e. $M_\phi = M^* = \frac{1}{2}(G_\theta + P)$, then optimization via Eq. (14) leads to minimization of $JSD(G\|P)$.*

*Proof.* Let $p$ denote the intermediate states. It would be used in the detailed proof. All we need to show is

$$\nabla_\theta \mathop{\mathbb{E}}_{s \sim G_\theta} \Big[ \sum_{t=1}^{n} \pi_g(s_t)^\top (\log \pi_m(s_t) - \log \pi_g(s_t)) \Big] \propto \nabla_\theta JSD(P\|G_\theta). \tag{15}$$

By inversely applying Lemma 2, the left part in Eq. (15) can be recovered as

$$\nabla_\theta \Big( \frac{1}{2} \mathop{\mathbb{E}}_{s \sim G_\theta} \Big[ \log \frac{G_\theta(s)}{M_\phi(s)} \Big] \Big), \tag{16}$$

which is equivalent to

$$\nabla_\theta \left( \mathop{\mathbb{E}}_{s \sim G_\theta} \Big[ \log \frac{G_\theta(s)}{M_\phi(s)} \Big] + \mathop{\mathbb{E}}_{s \sim P} \Big[ \log \frac{P(s)}{M_\phi(s)} \Big] \right). \tag{17}$$

Since now mediator is trained to be optimal, *i.e.* $M_\phi = M^*$, we have

$$\begin{aligned} (17) =& \nabla_\theta \left( \mathop{\mathbb{E}}_{s \sim G_\theta} \Big[ \log \frac{G_\theta(s)}{M^*(s)} \Big] + \mathop{\mathbb{E}}_{s \sim P} \Big[ \log \frac{P(s)}{M^*(s)} \Big] \right) \\ =& 2\nabla_\theta J\hat{S}D(P\|G_\theta) \propto \nabla_\theta J\hat{S}D(P\|G_\theta). \end{aligned} \tag{18}$$

This means training through CoT leads to minimization of $J\hat{S}D(P\|G_\theta)$. When the mediator is trained to be optimal, $J\hat{S}D(P\|G_\theta) = JSD(P\|G_\theta)$. This verifies the theorem.

### 3.4 DISCUSSION

#### 3.4.1 ADVANTAGES OVER PREVIOUS METHODS

CoT has several practical advantages over previous methods, including MLE, Scheduled Sampling (SS) (Bengio et al., 2015) and adversarial methods like SeqGAN (Yu et al., 2017).

First, although CoT and GAN both aim to optimize an estimated JSD, CoT is exceedingly more stable than GAN. This is because the two modules, namely generator and mediator, have similar tasks, *i.e.* to approach the same data distribution **generatively** and **predictively**. The superiority of CoT over inconsistent methods like Scheduled Sampling is obvious, since CoT theoretically guarantees the training effectiveness. Compared with methods that require pre-training in order to reduce variance like SeqGAN (Yu et al., 2017), CoT is computationally cheaper. More specifically, under recommended settings, CoT has the same order of computational complexity as MLE.

Besides, CoT works independently. In practice, it does not require model pre-training via conventional methods like MLE. This is the first time that unbiased unsupervised learning is achieved on sequential discrete data without using supervised approximation for variance reduction or sophisticated smoothing as in Wasserstein GAN with gradient penalty (WGAN-GP) (Gulrajani et al., 2017).

#### 3.4.2 THE NECESSITY OF THE MEDIATOR

An interesting problem is to ask why we need to train a mediator by mixing the samples from both sources $G$ and $P$, instead of directly training a predictive model $\hat{P}$ on the training set via MLE. There are basically two points to interpret this.

To apply the efficient training objective 13, one needs to obtain not only the mixture density model $M = \frac{1}{2}(P + G)$ but also its decomposed form in each timestep *i.e.* $M_\phi(s) = \prod_{t=1}^{n} M_\phi(s_t|s_{t-1})$, without which the term $\pi_m(s_t)$ in Eq 13 cannot be computed efficiently. This indicates that if we directly estimate $P$ and compute $M = \frac{1}{2}(G + P)$, the obtained $M$ will be actually useless since its decomposed form is not available.

Besides, as a derivative problem of "exposure bias", there is no guarantee for the model $\hat{P}$ to work well on the generated samples *i.e.* $s \sim G_\theta$ to guide the generator towards the target distribution. Given finite observations, the learned distribution $\hat{P}$ is trained to provide correct predictions for samples from the target distribution $P$. There is no guarantee that $\hat{P}$ can stably provide correct predictions for guiding the generator. Ablation study is provided in the appendix.

## 4 EXPERIMENTS

### 4.1 UNIVERSAL SEQUENCE MODELING IN SYNTHETIC TURING TEST

Following the synthetic data experiment setting in (Yu et al., 2017; Zhu et al., 2018), we design a synthetic Turing test, in which the negative log-likelihood $\text{NLL}_{oracle}$ from an oracle LSTM is calculated for evaluating the quality of samples from the generator. Particularly, to support our claim that our method causes little mode collapse, we calculated $\text{NLL}_{test}$, which is to sample an extra batch of samples from the oracle, and to calculate the negative log-likelihood measured by the generator. We show that under this more reasonable setting, our proposed algorithm reaches the state-of-the-art performance with exactly the same network architecture. Note that models like LeakGAN (Guo et al., 2017) contain architecture-level modification, which is orthogonal to our approach, thus will not be included in this part. The results are shown in Table 1.

#### 4.1.1 DISCUSSION

**Computational Efficiency** Although in terms of time cost per epoch, CoT does not achieve the state-of-the-art, we do observe that CoT is remarkably faster than previous RL-GAN approaches. Besides, consider the fact that CoT is a sample-based optimization algorithm, which involves time

Table 1: Likelihood-based benchmark and time statistics for synthetic Turing test. '-(MLE)' means the best performance is acquired during MLE pre-training.

| Model/Algorithm | $NLL_{oracle}$ | $NLL_{test}$(final/best) | best $NLL_{oracle+test}$ | time/epoch |
|---|---|---|---|---|
| MLE | 9.08 | 8.97/7.60 | 9.43 + 7.67 | **16.14 ± 0.97s** |
| SeqGAN (Yu et al., 2017) | 8.68 | 10.10/-(MLE) | (The same as MLE) | $817.64 \pm 5.41s$ |
| RankGAN (Lin et al., 2017) | 8.37 | 11.19/-(MLE) | (The same as MLE) | $1270 \pm 13.01s$ |
| MaliGAN (Che et al., 2017) | 8.73 | 10.07/-(MLE) | (The same as MLE) | $741.31 \pm 1.45s$ |
| Scheduled Sampling (Bengio et al., 2015) | 8.89 | 8.71/-(MLE) | (The same as MLE) | $32.54 \pm 1.14s$ |
| Professor Forcing (Lamb et al., 2016) | 9.43 | 8.31/-(MLE) | (The same as MLE) | $487.13 \pm 0.95s$ |
| CoT (ours) | **8.19** | **8.03/7.54** | **8.19 + 8.03** | $53.94 \pm 1.01s$ |

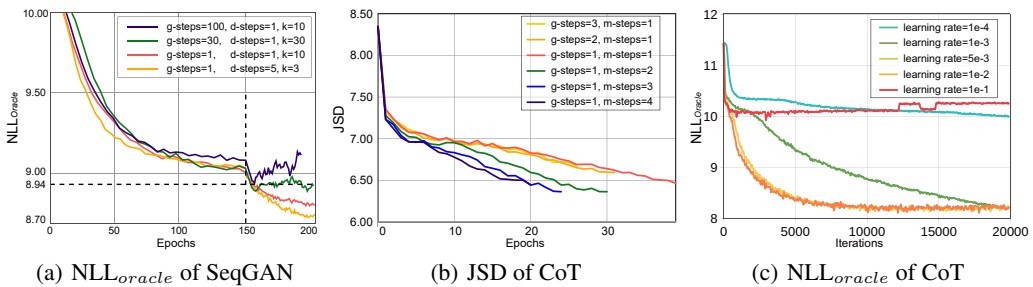

(a) $NLL_{oracle}$ of SeqGAN  (b) JSD of CoT  (c) $NLL_{oracle}$ of CoT

Figure 2: Curves of evaluation on JSD, $NLL_{oracle}$ during iterations of CoT under different training settings. To show the hyperparameter robustness of CoT, we compared it with the similar results as were evaluated in SeqGAN (Yu et al., 2017).

cost in sampling from the generator, this result is acceptable. The result also verifies our claim that CoT has the same order (*i.e.* the time cost only differs in a constant multiplier or extra lower order term) of computational complexity as MLE.

**Hyper-parameter Robustness.** We perform a hyper-parameter robustness experiment on synthetic data experiment. When compared with the results of similar experiments as in SeqGAN (Yu et al., 2017), our approach shows less sensitivity to hyper-parameter choices, as shown in Figure 2. Note that since in all our attempts, the evaluated JSD of SeqGAN fails to converge, we evaluated $NLL_{oracle}$ for it as a replacement.

**Self-estimated Training Progress Indicator.** Like the critic loss, *i.e.* estimated Earth Mover Distance, in WGANs, we find that the training loss of the mediator (9), namely *balanced NLL*, can be a real-time training progress indicator as shown in Figure 3. Specifically, in a wide range, balanced NLL is a good estimation of real $JSD(G\|P)$ with a steady translation, namely, $balanced\ NLL = JSD(G\|P) + H(G) + H(P)$.

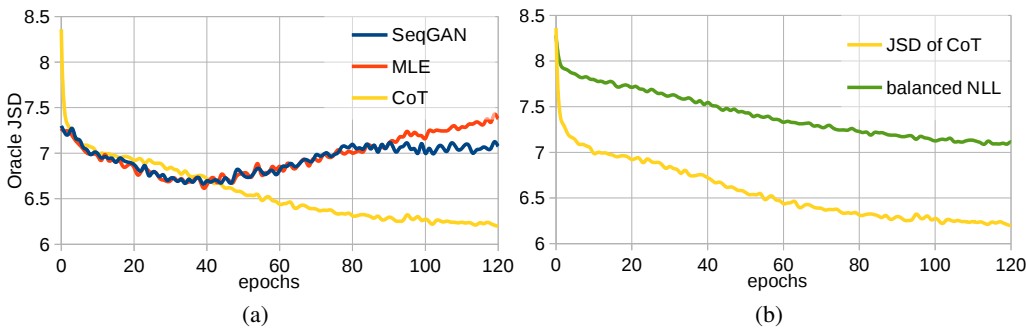

(a)  (b)

Figure 3: (a) Curves of training time $JSD(G\|P)$ for MLE, SeqGAN and CoT. (b) Curves of balanced NLL and real JSD. Both results are from synthetic data experiments. Note that balanced NLL is considered to have only a constant translation of the estimated JSD by the mediator.

Table 2: N-gram-level quality benchmark: BLEU on test data of EMNLP2017 WMT News

| Model/Algorithm | BLEU-2 | BLEU-3 | BLEU-4 | BLEU-5 |
|---|---|---|---|---|
| MLE | 0.781 | 0.482 | 0.225 | 0.105 |
| SeqGAN (Yu et al., 2017) | 0.731 | 0.426 | 0.181 | 0.096 |
| RankGAN (Lin et al., 2017) | 0.691 | 0.387 | 0.178 | 0.095 |
| MaliGAN (Che et al., 2017) | 0.755 | 0.456 | 0.179 | 0.088 |
| LeakGAN (Guo et al., 2017) | 0.835 | 0.648 | 0.437 | 0.271 |
| TextCoT-basic (ours) | 0.785 | 0.489 | 0.261 | 0.152 |
| TextCoT-strong (ours) | 0.800 | 0.501 | 0.273 | 0.200 |
| TextCoT-strong ($\alpha = 1.5$) (ours) | **0.856** | **0.701** | **0.510** | **0.310** |

Table 3: Diversity benchmark: estimated Word Mover Distance (eWMD) and $\text{NLL}_{test}$

| Model/Algorithm | $\text{eWMD}_{test}$ | $\text{eWMD}_{train}$ | $\text{NLL}_{test}$ |
|---|---|---|---|
| MLE | 1.015 ($\sigma = 0.023$) | 0.947 ($\sigma = 0.019$) | 2.365 |
| SeqGAN (Yu et al., 2017) | 2.900 ($\sigma = 0.025$) | 3.118 ($\sigma = 0.018$) | 3.122 |
| RankGAN (Lin et al., 2017) | 4.451 ($\sigma = 0.083$) | 4.829 ($\sigma = 0.021$) | 3.083 |
| MaliGAN (Che et al., 2017) | 4.891 ($\sigma = 0.061$) | 4.962 ($\sigma = 0.020$) | 3.240 |
| LeakGAN (Guo et al., 2017) | 1.803 ($\sigma = 0.027$) | 1.767 ($\sigma = 0.023$) | 2.327 |
| TextCoT-basic (ours) | **0.766** ($\sigma = 0.031$) | **0.886** ($\sigma = 0.019$) | 2.247 |
| TextCoT-strong (ours) | 0.923 ($\sigma = 0.018$) | 0.941 ($\sigma = 0.016$) | **2.144** |

## 4.2 TEXTCOT: ZERO-PRIOR LONG & DIVERSE TEXT GENERATION

As an important sequential data modeling task, zero-prior text generation, especially long and diversified text generation, is a good testbed for evaluating the performance of a generative model.

Following the experiment proposed in LeakGAN (Guo et al., 2017), we choose EMNLP 2017 WMT News Section as our dataset, with maximal sentence length limited to 51. We pay major attention to both **quality** and **diversity**. To keep the comparison fair, we present two implementations of CoT, namely CoT-basic and CoT-strong. As for CoT-basic, the generator follows the settings of that in MLE, SeqGAN, RankGAN and MaliGAN. As for CoT-strong, the generator is implemented with the similar architecture in LeakGAN.

For quality evaluation, we evaluated BLEU on a small batch of test data separated from the original dataset. For diversity evaluation, we evaluated the estimated Word Mover Distance (Kusner et al., 2015), which is calculated through training a discriminative model between generated samples and real samples with 1-Lipschitz constriant via gradient penalty as in WGAN-GP (Gulrajani et al., 2017). To keep it fair, for all evaluated models, the architecture and other training settings of the discriminative models are kept the same.

The results are shown in Table 2 and Table 3. In terms of generative quality, CoT-basic achieves state-of-the-art performance over all the baselines with the same architecture-level capacity, especially the long-term robustness at n-gram level. CoT-strong using a conservative generation strategy, *i.e.* setting the inverse temperature parameter $\alpha$ higher than 1, as in (Guo et al., 2017) achieves the best performance over all compared models. In terms of generative diversity, the results show that our model achieves the state-of-the-art performance on all metrics including $\text{NLL}_{test}$, which is the optimization target of MLE.

## 5 CONCLUSION

We proposed Cooperative Training, a novel training algorithm for generative modeling of discrete data. CoT optimizes Jensen-Shannon Divergence, which does not have the *exposure bias* problem as the forward KLD. Models trained via CoT shows promising results in sequential discrete data modeling tasks, including sample quality and the generalization ability in likelihood prediction tasks.

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

## A    Detailed Derivation of the Algorithm

$$(10) = \nabla_\theta \left( -\frac{1}{2} \mathop{\mathbb{E}}_{s_t \sim G_\theta} \left[ \log G_\theta(s_t) - \log M_\phi(s_t) \right] \right)$$

$$= \nabla_\theta \left( -\frac{1}{2} \mathop{\mathbb{E}}_{s_t \sim G_\theta} \left[ \frac{G_\theta(s_{t-1}) G_\theta(s_t|s_{t-1})}{G_\theta(s_t)} (\log G_\theta(s_t) - \log M_\phi(s_t)) \right] \right)$$

$$= \nabla_\theta \left( -\frac{1}{2} \mathop{\mathbb{E}}_{s_t \sim G_\theta} \left[ \frac{G_\theta(s_{t-1}) G_\theta(s_t|s_{t-1})}{G_\theta(s_t)} \left( \log G_\theta(s_t|s_{t-1}) G_\theta(s_{t-1}) - \log M_\phi(s_t|s_{t-1}) M_\phi(s_{t-1}) \right) \right] \right)$$

$$= -\frac{1}{2} \nabla_\theta \left( \sum_{s_t} G_\theta(s_{t-1}) G_\theta(s_t|s_{t-1}) \left( \log G_\theta(s_t|s_{t-1}) - \log M_\phi(s_t|s_{t-1}) \right) \right.$$
$$\left. + \sum_{s_t} G_\theta(s_{t-1}) G_\theta(s_t|s_{t-1}) \log \frac{G_\theta(s_{t-1})}{M_\phi(s_{t-1})} \right)$$

$$= -\frac{1}{2} \nabla_\theta \left( \sum_{s_t} G_\theta(s_{t-1}) G_\theta(s_t|s_{t-1}) \left( \log G_\theta(s_t|s_{t-1}) - \log M_\phi(s_t|s_{t-1}) \right) \right.$$
$$\left. + \sum_{s_{t-1}} \left( G_\theta(s_{t-1}) \log \frac{G_\theta(s_{t-1})}{M_\phi(s_{t-1})} \right) \sum_{s_t} G_\theta(s_t|s_{t-1}) \right) \quad \text{(here } s_{t-1} \text{ iterates over all prefixes of the sequences in } \{s_t\})$$

$$= -\frac{1}{2} \nabla_\theta \left( \sum_{s_t} G_\theta(s_{t-1}) G_\theta(s_t|s_{t-1}) \left( \log G_\theta(s_t|s_{t-1}) - \log M_\phi(s_t|s_{t-1}) \right) + \sum_{s_{t-1}} G_\theta(s_{t-1}) \log \frac{G_\theta(s_{t-1})}{M_\phi(s_{t-1})} \right)$$

$$= -\frac{1}{2} \nabla_\theta \left( \sum_{s_t} G_\theta(s_{t-1}) G_\theta(s_t|s_{t-1}) \left( \log G_\theta(s_t|s_{t-1}) - \log M_\phi(s_t|s_{t-1}) \right) + \mathop{\mathbb{E}}_{s_{t-1} \sim G_\theta} \left[ \log \frac{G_\theta(s_{t-1})}{M_\phi(s_{t-1})} \right] \right)$$

$$= -\frac{1}{2} \nabla_\theta \left( \sum_{s_{t-1}} G_\theta(s_{t-1}) \sum_{s_t} G_\theta(s_t|s_{t-1}) \left( \log G_\theta(s_t|s_{t-1}) - \log M_\phi(s_t|s_{t-1}) \right) + \mathop{\mathbb{E}}_{s_{t-1} \sim G_\theta} \left[ \log \frac{G_\theta(s_{t-1})}{M_\phi(s_{t-1})} \right] \right)$$

$$= (12)$$

## B    Sample Comparison and Discussion

Table 4 shows samples from some of the most powerful baseline models and our model.

Observation of the model samples indicates that:

- CoT produces remarkably more diverse and meaningful samples when compared to Leak-GAN.
- The consistency of CoT is significantly improved when compared to MLE.

## C    Further Discussions about the Experiment Results

**The Optimal Balance for Cooperative Training**  We find that the same learning rate and iteration numbers for the generator and mediator seems to be the most competitive choice. As for the architecture choice, we find that the mediator needs to be slightly stronger than the generator. For the best result in the synthetic experiment, we adopt exactly the same generator as other compared models and a mediator whose hidden state size is twice larger (with 64 hidden units) than the generator.

Theoretically speaking, we can and we should sample more batches from $G_\theta$ and $P$ respectively for training the mediator in each iteration. However, if no regularizations are used when training the mediator, it can easily over-fit, leading the generator's quick convergence in terms of $KL(G_\theta \| P)$ or $\text{NLL}_{oracle}$, but divergence in terms of $\widehat{JSD}(G_\theta \| P)$. Empirically, this could be alleviated by applying dropout techniques (Srivastava et al., 2014) with 50% keeping ratio before the output layer of RNN. After applying dropout, the empirical results show good consistency with our theory that, more training batches for the mediator in each iteration is always helpful.

However, applying regularizations is not an ultimate solution and we look forward to further theoretical investigation on better solutions for this problem in the future.

Table 4: WMT News Samples from Different Models

| Sources | Example |
|---|---|
| LeakGAN | (1) It's a big advocate for therapy is a second thing to do, and I'm creating a relationship with a nation.
(2) It's probably for a fantastic footage of the game, but in the United States is already time to be taken to live.
(3) It's a sad House we have a way to get the right because we have to go to see that, " she said.
(4) I'm not sure if I thank a little bit easier to get to my future commitment in work, " he said.
(5) " I think it was alone because I can do that, when you're a lot of reasons, " he said.
(6) It's the only thing we do, we spent 26 and $35(see how you do is we lose it," said both sides in the summer. |
| CoT | (1) We focus the plans to put aside either now, and which doesn't mean it is to earn the impact to the government rejected.
(2) The argument would be very doing work on the 2014 campaign to pursue the firm and immigration officials, the new review that's taken up for parking.
(3) This method is true to available we make up drink with that all they were willing to pay down smoking.
(4) The number of people who are on the streaming boat would study if the children had a bottle - but meant to be much easier, having serious ties to the outside of the nation.
(5) However, they have to wait to get the plant in federal fees and the housing market's most valuable in tourism. |
| MLE | (1) after the possible cost of military regulatory scientists, chancellor angela merkel's business share together a conflict of major operators and interest as they said it is unknown for those probably 100 percent as a missile for britain.
(2) but which have yet to involve the right climb that took in melbourne somewhere else with the rams even a second running mate and kansas.
(3) " la la la la 30 who appeared that themselves is in the room when they were shot her until the end " that jose mourinho could risen from the individual .
(4) when aaron you has died, it is thought if you took your room at the prison fines of radical controls by everybody, if it's a digital plan at an future of the next time. |

**Possible Derivatives of CoT** The form of equation 13 can be modified to optimize other objectives. One example is the backward KLD (*a.k.a.* Reverse KLD) *i.e.* $KL(G\|P)$. In this case, the objective of the so-called "Mediator" and "Generator" thus becomes:

"Mediator", now it becomes a direct estimator $\hat{P}_\phi$ of the target distribution $P$:

$$J_{\hat{p}}(\phi) = \mathbb{E}_{s\sim P}[-\log(\hat{P}_\phi(s))]. \tag{19}$$

Generator:

$$\nabla_\theta J_g(\theta) = \nabla_\theta \mathbb{E}_{s\sim G_\theta}\Big[\sum_{t=0}^{n-1} \pi_g(s_t)^\top (\log \pi_{\hat{p}}(s_t) - \log \pi_g(s_t))\Big]. \tag{20}$$

Such a model suffers from so-called mode-collapse problem, as is analyzed in Ian's GAN Tutorial (Goodfellow, 2016). Besides, as the distribution estimator $\hat{P}\phi$ inevitably introduces unpredictable behaviors when given unseen samples *i.e.* samples from the generator, the algorithm sometimes fails (numerical error) or diverges.

In our successful attempts, the algorithm produces similar (not significantly better than) results as CoT. The quantitive results are shown as follows:

Table 5: N-gram-level quality benchmark: BLEU on test data of EMNLP2017 WMT News (New Split)

| Model/Algorithm | BLEU-2 | BLEU-3 | BLEU-4 | BLEU-5 | eWMD |
|---|---|---|---|---|---|
| CoT-basic (ours) | 0.850 | 0.571 | 0.316 | 0.169 | **1.001** ($\sigma = 0.020$) |
| Reverse KL (ours) | **0.860** | **0.590** | **0.335** | **0.181** | 1.086 ($\sigma = 0.014$) |

Although under evaluation of weak metrics like BLEU, if successfully trained, the model trained via Reverse KL seems to be better than that trained via CoT, the disadvantage of Reverse KL under evaluation of more strict metric like eWMD indicates that Reverse KL does fail in learning some aspects of the data patterns *e.g.* completely covering the data mode.

