# OpenReview forum: "CoT: Cooperative Training for Generative Modeling of Discrete Data"
_ICLR.cc/2019/Conference_

### Official Review · AnonReviewer1 · 2018-10-23
**nice idea**

**Rating:** 6
**Confidence:** 4

**Review:**

Pros:
This paper is easy to follow. The idea is nice in three folds.
1. By changing the auxiliary model's role from a discriminator to a mediator, it directly optimizes the JSD measure, which is a symmetrized and smoothed version of KL divergence.
2. Moreover, the mediator and the generator follow similar predictive goals, rather than the opposite  goals of G and D in GANs.
3. For discrete sequential data, it avoids approximating expected rewards using Markov rollouts.

Cons:
Some details are missing in the experiments.
1. In Table 2 of [A], LeakGAN, SeqGAN and RankGAN all show significantly better performances in terms of BLEU on EMNLP2017 WMT, compared to results reported in Table 3 of the submission. Any difference?
2. The Word Mover Distance is computed by training a discriminator, which could be unstable. Could you provide other metrics to evaluate diveristy like self-bleu?

[A] Guo, Jiaxian, et al. "Long text generation via adversarial training with leaked information." arXiv preprint arXiv:1709.08624 (2017).

Misc:
1. How will the number of samples (i.e. batch size) affect CoT ?
2. How is the applicability of CoT for continuous data? It seems to me there is no theoretical difficulties to apply CoT on continuous data.

---

> ### Author Response · Authors · 2018-11-07
> **Reply to ICLR AnonReviewer1**
>
> Thanks for reviewing our paper!
>
> Response to your concerns:
>
> 1. We have contacted one of the authors of the LeakGAN, finding that the data pre-processing and post-processing of ours and theirs are different. This makes the results quite different.
> 2. We will present with an error bar in the coming revised version.
>
> Response to Misc:
>
> 1. This is an interesting topic, we would have some discussion about it if we have found interesting conclusions.
>
> 2. We've actually implemented a continous version of CoT, of which the prior distribution is replaced by Beta Distribution instead of Multinomial Distribution in the current discrete version of CoT. However such a model does not perform well. This is an interesting direction for further research and survey.

---

> > ### Comment · AnonReviewer1 · 2018-12-14
> > **Response to rebuttal**
> >
> > Generally the reply responses my concerns.
> > I found the authors' comments towards other non-official reviewers' questions are very useful and important, including points that I have not considered at the first place. Some questions are well handled.
> > 1. it seems most people have questions towards the setting of experiments, so maybe some clearance is needed.
> > 2. the question and response on "why reinforce is not needed" is important and should be included in the final version.
> > 3. also the question and response on "what if the mediator fails to learn optimally".
> >
> > Overall, I think the authors well respond to others' concerns. But I think some content needs to be included in the final version. To encourage the authors add these content, I may slightly lower my rating.

---

> > > ### Author Response · Authors · 2018-12-14
> > > **Response to response**
> > >
> > > Thank you for your advice.
> > >
> > > 1. The paper was revised in the revision period to address most of the comments on experiments.  We consider it now to be more solid against the previous comments and criticisms. You may want to re-check it.
> > >
> > > 2. We agree, as we promise the corresponding discussion would be included in the final version of the paper.
> > >
> > > 3. Actually, the experiment of robustness (see Sec 4.1.1) shows that in pracice, whether the mediator is trained to be optimal is not so important. See Fig2(b) and its legend. We state that the algorithm still works well under a variety of g-m balancing settings. In the extreme case, even if the mediator is remarkably less trained (g-steps=3, m-steps=1), the algorithm still works well and suceeds to converge.
> > >
> > > We appreciate your comments, as we promise that these content would definitely be added to the final version of the paper.

---

### Official Review · AnonReviewer2 · 2018-10-31
**Interesting and promising method for generative modelling of sequence data without policy gradient**

**Rating:** 7
**Confidence:** 2

**Review:**

The paper proposes an interesting method, where the discriminator is replaced by a component that estimates the density that is the mixture of the data and the generator's distributions. In a sense, that component is only a device that allows estimating a Jensen-Shannon divergence for the generator to then be optimized against. Other GAN papers have replaced their discriminator by a similar device (e.g., WGANs, ..), but the present formulation seems novel. The numerical experiments presented on a synthetic Turing test and text generation from EMNLP's 2017 news dataset appear promising.

Overall, the mediator seems to allow to achieve lower Jensen-Shannon (JS) divergence values in the experiments (and is kind of designed for that). Although this may be an improvement with respect to existing methods for discrete sequential data, it may also be limited in that it may not easily extend to other types of divergences that have proved superior to JS in some continuous settings.

The paper is rather clear, although there are lots of small grammatical errors as well as odd formulations which end up being distracting or confusing. The language should be proof-read carefully.

Pros:
- Generative modeling of sequence data still in its infancy
- Potentially lower variance than policy gradient approaches
- Experiments are promising

Cons:
- Lots of grammatical errors and odd formulations

Questions:
- Equation 14: what does it mean to find the "maximum entropy solution" for the given optimization problem?
- Figure 2: how do (b) and (c) relate to each other?

Remarks, small typos and odd formulations:
- "for measuring M_\/phi": what does measuring mean in this context?
- What does small m refer to? Algorithm 1 says the total number of steps  but it is also used in the main text as an index for J and \pi (for mediator?)
- Equation block 8: J_m has not been defined yet
- "the supports of distributions G and P"... -> G without subscript has now been defined in this context
- "if the training being perfect"
- "tend to get stuck in some sub-optimals"
- the learned distribution "collapseS"
- "since  the data distribution is, thus ..."
- "that measures a" -> "that estimates a ..."?
- "a predictive module": a bit unclear - generative v. discriminative is more usual terminology
- "is well ensured"
- "with the cost of diversity" -> "at the cost of diversity"?
- "has theoretical guarantee"
- in the references: "ALIAS PARTH GOYAL" (all caps)
- "let p denote the intermediate states": I don't understand what this is. Where is "p" used? (proof of Theorem 3)
- "CoT theoretically guarantees the training effectiveness": what does that mean?
- Figure 3: "epochs" -> "Epochs"
- Algorithm 1: what does "mixed balanced samples" mean? Make this more precise
- "wide-ranged"
- Equation 10 is too long and equation number is not properly formatted
- Figures hard to read in black & white
- Figure 2 doesn't use the same limits for the Y axis of the two NLL plots, making comparisons difficult. The two NLL plots are also not side-by-side

---

> ### Author Response · Authors · 2018-11-07
> **Response to AnonReviewer2**
>
> Thanks for reviewing our paper!
>
> Response to your concerns:
>
> We will provide with a carefully-revised version of the paper according to your generous suggestions.
>
> Answer to the questions:
> 1. If there is no constraint on the entropy of the solution of the objective function, the objective would not be equivalent to minimization of JSD. Instead, it would simply be calculating the entropy of M, which is useless.
>
> 2. Figure 2 (a)(b) shows CoT is more robust under its main evaluation to g-m balance compared to the g-d balance of SeqGAN. Ideally, Figure 2(a) should also be showing SeqGAN's performance under evaluation of JSD, however, in our attempts, SeqGAN always diverges under such evaluation. Figure 2 (c) show that the convergence of CoT is steady and quite fast under evaluation of NLL_{oracle}, which is biased on quality.

---

### Official Review · AnonReviewer3 · 2018-11-06
**Original idea, clear presentation**

**Rating:** 7
**Confidence:** 2

**Review:**

*Summary*
A clear an interresting presentation on learning sequences distributions. It achieve this objective by replacing the discriminator with a "mediator", a mixture between the training distribution and the target distribution which is estimated via maximum likelihood.

*Pros*
- Original idea for modelling distribution of sequence data
- Theoretical convergence in the Jensen Shanon divergence sense
- Promising experiments

*Cons*
- No major cons to the best of my knowledge

*Typos*
- It would be very nice to have black and white / color blind friendly graphs
- Eq 10 too long
- Introduce J_m & J_g in  sentence
- Coma at the end of Eq 5, and maybe align Generator and Discriminator in some position (e.g. at the semi colon).
- missing dot at Eq 8.

*Question*
- How would you ensure reproducibility (e.g. link to some code?)
- Is there any hope to obtain consistency (convergence) wrt other metrics?

---

> ### Author Response · Authors · 2018-11-07
> **Response to AnonReviewer3**
>
> Thanks for reviewing our paper.
>
> 1. For reproducibility, we are preparing for a open-source code base. After the paper is de-anonymized, we will attach a link to it.
>
> 2. Before the paper of CoT is completed, we have had attempts at several different divergences, including JSD(CoT), Reverse KL(as is described in the appendix), Wasserstein-1 distance, etc. However, only CoT and Reverse KL succeed in getting rid of pre-training via MLE. Reverse KL appears to have mode collapsing problem, therefore CoT is finally the chosen model. However, this is a good direction for further research. We are also interested.

---

> > ### Comment · AnonReviewer3 · 2018-11-20
> > **Thank you**
> >
> > Thank you for the reply!

---

### Public Comment · (anonymous) · 2018-09-28
**Table 2 and 3 are misleading!**

Table 2 shows 'quality' performance. By decreasing the temperature (decreased entropy) of your model you can outperform your baselines. However, in Table 3 you present 'diversity' performance but you don't report your model at the lower temperature (\alpha=1.5) and this model has to do worse because of the quality/diversity trade-off.

Thus, there is no way to know if CoT is state-of-the-art or for that manner actually outperforms any other model.

---

> ### Author Response · Authors · 2018-09-29
> **Table 2 and 3 are designed as such to make fair comparison with LeakGAN**
>
> Thank you for your attention! As we would like to point out, there is one thing special about LeakGAN when compared to other baseline models.  In LeakGAN's default settings,when it is used to generate samples, the temperature of the generator is adjusted to be a little bit lower (i.e. \alpha = 1.5). However, when LeakGAN is used to compute the predicted NLL, \alpha will be set to 1.0. To keep the comparison fair, we show results of CoT-strong under different settings (\alpha = 1.0 and \alpha = 1.5) to support our claim that:
>
> 1. When we set \alpha = 1.0 i.e. keep the temperature parameters as they originally are, CoT-basic and CoT-strong outperforms baseline models with the same settings, including MLE, SeqGAN, RankGAN and MaliGAN.
>
> 2. When \alpha is set to 1.5, CoT-strong outperforms LeakGAN.
>
> The proposed CoT does reach the state-of-the-art, since in both cases CoT reaches the state-of-the-art. As for diversity benchmarks, since every evaluated model sets \alpha to 1.0 in this case, such classified discussion is not necessary.

---

> > ### Public Comment · (anonymous) · 2018-11-05
> > **Still Missing Critical Component for Table 2**
> >
> > Hi,
> >
> > To make a more fair comparison for Table 2, it's better to first set \alpha = 1.0 for all models (including Leakgan) and compare them with CoT. Then set \alpha = 1.5 for all models and compare the results again.
> >
> > Basically you need to show that your model outperforms Leakgan under both settings (\alpha = 1.0 and \alpha = 1.5) to claim that the new model is state-of-art. Since you have compared Leakgan's NLL under \alpha = 1.0, why don't you also record the BLEU score and compare it with CoT under \alpha = 1.0?

---

> > > ### Author Response · Authors · 2018-11-05
> > > **Please refer to LeakGAN's original paper**
> > >
> > > Thanks for your attention. We would like to explain our considerations when we were writting this part. In the paper of LeakGAN, as is proposed by the authors, a typical setting of LeakGAN shall be:
> > >
> > > 1. When used for generative purposes, alpha is always set to be 1.5 (as they said ``conservative strategy'').
> > > 2. When used for evaluating NLL or sampling trajectories for reinforcement learning, alpha is always set to be 1.0.
> > >
> > > In other words, in LeakGAN's original paper, they do not guarantee the model would also perform well under the settings as you've described.
> > >
> > > However, the authors of LeakGAN did update a new version in their official github code base, where the temperature trick is completely removed. We would update a new version with related results when revision is enabled. Thank you.

---

### Public Comment · (anonymous) · 2018-10-16
**Are pad tokens included in the calculation of NLL_{test} (Table 3) ?**

Hi,

I am somewhat familiar with the EMNLP2017 WMT News dataset, and the NLL values reported are quite lower than what I'm used to seeing. It is possible that the PAD tokens are included in the calculation of the likelihood ?

Thank you

---

> ### Author Response · Authors · 2018-10-17
> **PAD tokens are included.**
>
> Hi,
>
> For all evalutated models, the padding tokens are included in the calculation of NLL. The reason of doing so is that we observed some model incorrectly generate non-padding tokens even if the model has generated a few padding tokens. The reason may be that in text generation tasks, generated tokens are SAMPLED from each time's probability prediction over the vocabulary instead of directly MAGINALIZED to be the argmax token. This, however, could be considered as an important feature when evaluating different training algorithms.
>
> Example:
> One typical failure:
>
> A cat is sleeping on the table . <PAD> <PAD> with <PAD> <PAD> <PAD> . <PAD> ......
>
> To penalize such failures, when calculating the NLL, all paddings are not eliminated. We hope our such consideration makes sense to you.

---

> > ### Public Comment · (anonymous) · 2018-10-17
> > **Why on the test set?**
> >
> > Hi,
> >
> > thanks for the answer. While I understand why you could need this for training, why not mask the tokens when calculating the likelihood on the test set ?  One could have arbitrarily good NLL performance by simply increasing the sentence length with PAD tokens, since the distribution becomes fully deterministic after the first PAD token.
> >
> > It would be greatly appreciated if, for reproducibility purposes, you could update the paper with the correct likelihood. This will allow other researchers to correctly compare their model with yours.
> >
> > Thank you

---

> > > ### Author Response · Authors · 2018-10-17
> > > **PAD tokens should be included in this case even in testing the NLL**
> > >
> > > Hi,
> > >
> > > What we are trying to say is that even if when evaluting the test NLL, pad tokens should not be eliminated so that any unstability of keeping the padding segment consistent (i.e. the predicted likelihood of pad token should be almost always 1.0 since the first pad token is generated) would be detected and penalized by doing so. Such setting does not introduce any unfairness of the comparison, since for all evaluated models, the padded sequence length (max sequence length) is the same. One could not have arbitrarily good NLL performance by doing what you've described, since in our setting it is not allowed to place additional <PAD> after the sequence (otherwise it makes the padded sequence longer than the maximum sequence length limit).
> > >
> > > Here is an example of our consideration:
> > >
> > > Suppose there are two evaluated models, namely A and B.
> > >
> > > Evaluated Sequence: I have a pen . <PAD> <PAD> <PAD>
> > >
> > > Probability prediction of each step:
> > >      I      have     a     pen    .       <PAD>   <PAD>  <PAD>  <PAD>
> > > A: 0.3    0.5     0.1   0.5    0.7       0.99      0.8         0.999   0.9999
> > > B: 0.3    0.5     0.1   0.5    0.7       0.99      0.999     0.999   0.9999
> > >
> > > We prefer model B, since if the prefix "I have a pen . <PAD>" is given, model B would have almost 1.0 probability to generate a correctly padded sequence, while for model A it would have only 0.8 probability to do so, even if for non-padding part the two models are actually the same.
> > >
> > > As for making it easier for other researchers to make correct comparison, we recommend implementing and evaluating with Texygen, which can automatically deal with these data preprocessing issues.
> > >
> > > Best

---

> > > > ### Public Comment · (anonymous) · 2018-10-17
> > > > **Reproducibility is impossible with such an approach**
> > > >
> > > > Hi,
> > > >
> > > > thanks again for your fast response. While I agree that the comparison is fair across all models using the same - fixed - sequence length, my issue is the following:
> > > >
> > > > In the Texygen repo you mentioned, the NLL values are calculated as :
> > > >
> > > >         NLL = -tf.reduce_sum(
> > > >             tf.one_hot(tf.to_int32(tf.reshape(self.x, [-1])), self.num_vocabulary, 1.0, 0.0) * tf.log(
> > > >                 tf.clip_by_value(tf.reshape(self.g_predictions, [-1, self.num_vocabulary]), 1e-20, 1.0)
> > > >             )
> > > >         ) / (self.sequence_length * self.batch_size)
> > > >
> > > > On the last line, we see that we are dividing by the sequence length. Therefore, by making the sentences arbitrarily large with <PAD> tokens, NLL_{test} becomes increasingly better, since predicting <PAD> tokens is trivial. Therefore, one can see that as self.sequence_length --> infinity , NLL --> 0.
> > > >
> > > > The issue is that it is hard, if not impossible to replicate the results obtained in your paper without knowing the value of self.sequence_length. Therefore, benchmarking against your algorithm on NLL_{test} is unnecessarily hard. Results presented should be agnostic of such preprocessing details. Moreover, on a more personal note, Texygen has some major design flaws, such as constantly dumping words to text files instead of simply keeping then on GPU (or RAM for that matter), making it somewhat painful tool for research. Avoiding having to using Texygen to reproduce results would be great.
> > > >
> > > > Thank you

---

> > > > > ### Author Response · Authors · 2018-10-17
> > > > > **Sequence length information has been already provided, with which reproducibility is actually possible**
> > > > >
> > > > > If you pay attention to the paper, we've actually provided with data pre-processing details (see page 8 Sec 4.2). The sequence length limit is set to be 51.

---

> > > > > > ### Public Comment · (anonymous) · 2018-10-17
> > > > > > **Awesome**
> > > > > >
> > > > > > Great! Thanks you kindly

---

### Public Comment · (anonymous) · 2018-10-30
**question relates to "Detailed Derivation of The Algorithm"**

for line 2 within the proof,  G[S_(t-1)]G[S_(t)|S_{t-1}] = G[S_{t}] seems to be wrong. It should be SUM(i) G[S_(t-1)^{i}]G[S_(t)|S_{t-1}^{i}] = G[S_{t}]

---

> ### Author Response · Authors · 2018-10-30
> **Please refer to our notations**
>
> Please refer to our notations at Sec 2.
> In this case, for any given sequence S_t, its t-1-length prefix S_{t-1} is unique.

---

### Public Comment · (anonymous) · 2018-11-07
**major concerns with algorithm and evaluation (1/2)**

Disclaimer: I reviewed this paper for a previous conference. The authors have not updated the paper to address any of my concerns, so I have copied my review below.

This paper presents a new technique (CoT) for training gen. models with tractable densities on discrete data. Instead of minimizing the KL divergence as in MLE, CoT minimizes the JS divergence. Unlike GANs that learn a critic to approximate the density ratio between the generated and true samples, CoT learns a “mediator” that approximates the mixture of the two densities, and uses that to construct the density ratio. They evaluate the technique on a synthetic sequence dataset from SeqGAN, and the EMNLP 2017 WMT News Section, and show improved performance in terms of sample quality (oracle NLL, BLEU) and sample diversity (test NLL, word mover distance).

Overall, I found the idea of using a mediator to construct the density ratio interesting, but the use of an extra mediator is not necessary (see below).  The core claim that the technique is “unbiased, low-variance, and computationally efficient” is not sufficiently demonstrated. In particular, computing gradients of the objective function still require REINFORCE to differentiate through the sampling process, and the technique requires training an additional tractable density model on the mixture. Furthermore, there are no samples in the paper, and to achieve improved sample quality in terms of BLEU the authors adjust the temperature. Based on the experimental results, it’s not obvious that this set of techniques improves performance over the MLE baseline. The presented technique is also substantially more expensive as you have to sample from an autoregressive model at training time, and evaluate two densities instead of one (mediator and generative model). An improved version of the paper should clarify how gradients are computed efficiently, clean up the language and presentation throughout,  and present a more thorough evaluation of the technique versus MLE.

Major comments:
There is no reason to continually train and update an extra network to be the mediator. If you first train a network M with MLE, then that represents the best approximation in the model class to the data density, P. You can then create a new network G, and use G / ((P + G)/2.0) to construct the desired density ratio. This approach would require training M first, but removes the need for an explicit mediator network while training.

Paper is missing references to related work that leverages a tractable density as noise to estimate another density, e.g. noise contrastive estimation Gutmann & Hyvarinen 2010, b-GAN Uehara et al., 2017, and especially “On the distinguishability criteria for estimating generative models” Goodfellow 2015. The Goodfellow paper presents a technique they coin self-contrastive estimation which seems related to CoT.

While Theorem 3 makes sense to me, Theorem 4 is not obvious. The proof (for both theorems?) shows that the *value* of the CoT loss equals the JSD, but it does not prove that the *gradients* of the CoT loss equals the gradients of the JSD. To prove this, you need to invoke Danskin’s theorem. A simple example for why objective being equal at a point does not imply gradients are equal:
Let f(x) = x**2
Let m = x, and define g(x) = x * m
Then f(x)= g(x) at x=m, but the gradient of f(x) at m is 2*m, while the gradient of g(x) at m is just m.

The objective and gradient presented in eqs 13 and 14 have a gradient w.r.t. the parameters of the generative model \theta outside of an expectation over discrete samples s ~ G_\theta. The paper does not explain how you compute this gradient! To form an unbiased approximation of this expectation, you would have to use REINFORCE. How did you actually train these models? Do you use REINFORCE? Do you ignore this term in the gradient? I can’t see how to compute this without getting high bias or high variance gradients.

All the theory assumes the discriminator is trained to optimality. If is not, how does the technique fail? If the mediator can’t model (P+G), then your estimate of the density ratio will be wrong, and all the guarantees go out the window. If the mediator can model (P+G), then it can likely also model P, and then you can just build a perfect model with MLE.

There are no samples presented for the News Section (or synthetic) experiments! If the argument is that sample quality is improved, it would be great to have examples. The CoT-basic model performs almost the same as MLE! The CoT-strong model doesn’t present much of an improvement except when you use a different temperature for sampling. It looks like just using the strong model trained with MLE and tuning the temperature could achieve similar results.

There are a number of grammatical errors throughout, and the text is often confusing and unclear.

---

> ### Public Comment · (anonymous) · 2018-11-07
> **major concerns with algorithm and evaluation (2/2)**
>
> Minor comments:
> - The introduction does not sufficiently motivate the problem. What’s broken about MLE and why do we need to target different divergences or alter the training procedure?
> - non-standard notation for the x_t/s_t, maybe use s_{:t}
> - why is finite observations problematic for MLE? It is the only divergence that depends only on samples from the data distribution and not evaluating the data density.
> - Why does an ideal solution have to be symmetric or smooth? You should discuss the issues with KL and JSD when the distributions do not have overlapping support which is core to the Arjovsky and Bouttou paper. Also if the generative model G(\theta) is sufficiently powerful, then you can get good quality and diversity from any divergence.
> - The presentation of the generator loss in eqn 5 is confusing. By analogy to the original GAN, I expected the generator loss to be -E_{s ~ G}[log(1-D(s))], but you’re using the raw D(s) in this expression. This loss looks somewhat like REINFORCE with D(s) as a reward. Are you also doing a new rollout starting from each timestep t? I guess this is taken from SeqGAN but the objective is confusing to me and not presented clearly.
> - Again, I don’t see why a symmetric divergence are needed to give you quality and diversity.
> - when the mediator is not optimal (which it never will be in practice), CoT does not provide an unbiased mechanism for training with the JSD.
> - if the mediator M does not equal 0.5 * (P + G), then the estimate of JS will be biased, as will the gradients. Furthermore, you are ignoring one of the gradient terms when you are plugging into estimate JS: dL/dM dM / dG * dG/ d\theta. Yes, the quantities are equal, but the gradients are not necessarily equal (eq 18)
> - what do you mean by "generatively and predictively”? In practice, CoT is not consistent unless the mediator is exact, so I don’t think it’s fair to highlight consistency vs. scheduled sampling.
> - “same order of computational complexity as MLE”: isn’t it twice as expensive? And critically, you have to sample from the autoregressive generative model which will make it even more expensive.
> - this is a good point that you don’t require MLE pretraining, but a subroutine is training a model with MLE on data samples (mediator)
> -  This is a critical section highlighting why the mediator is necessary, but I have no idea what this paragraph is saying.
> - you should describe the synthetic data experiment so the paper is self-contained.
> Table 1: Please add error bars. Did you try early stopping with MLE?
> Figure 2: Would be interesting to plot these learning curves for MLE as well.
> Figure 3: Why not plot estimate of JSD using the current mediator? Does the estimate of JSD track the true JSD?
> Table 2: MLE baseline and CoT-basic performance look about the same. w/o error bars unclear if these are significant differences. To get good samples in terms of BLEU you adjust temperature. What happens if you do that with an MLE model?

---

> ### Author Response · Authors · 2018-11-07
> **Reply to "major concerns with algorithm and evaluation"**
>
> Thanks for your attention!
>
> We have seen your review before when submitted to a previous conference and here we would like to make an official response to it.
>
> In case you haven't noticed, we want to emphasize that this version of the paper is quite different from the version you've already seen, especially in the parts you've mentioned.
>
> Response to the first part of the review
>
> The reviewer makes the claims that using the mediator is not necessary and the proposed algorithm still incorporates REINFORCE algorithm. According to our submitted version of the paper, however, our presented algorithm disagrees with either of the claims.
>
> We are confused why the reviewer claims so since one of the most important idea we want to present in this paper is HOW TO avoid incorporating REINFORCE. Please refer to Eq.13, which is the key to the success of this. The paragraphs around Eq.13 describe our approach in details.
>
> For the necessity of the mediator, please check Sec 3.4.2, which is quite different from the version you have read. Your suggested version of the model may not be practically implementable because it cannot provide a probability prediction IN EACH TIMESTEP, which is very important if this module is to be used by CoT. Notice that M(x|s_t) is not equal to (G(x|s_t) + P(x|s_t)) / 2, making the factorization actually non-trivial.
>
> Response to the second part of the review
>
> Before this version of the paper is submitted to ICLR, we have deleted Theorem 4 in an earlier version of the paper as we consider your concerns about it to be correct. (thanks!) In our current version, we treat Mediator as an IMPLICIT estimator of JSD. The reason why such estimation is implicit is that the calculation of entropy of the input data is non-trivial. However, if you pay attention to Figure 3 and the paragraphs around it, we've shown balanced NLL, which, in theory, is actually only different in a constant from the estimated JSD. In practice, as the model's estimation of data entropy also improves, such difference may also change steadily as the training proceeds.
>
> About the behavior of the model when the mediator is not trained to optimality, we have empirically shown that it is still stable. Note that GANs also do not have such a guarantee, and in practice it is much more unstable than our approach. Please refer to Figure 2(b).
>
>
> We have collected samples from three typical models and shown them in the appendix. Please also check it.
>
> Response to minor comments:
>
> We appreciate your suggestions. However, as the paper has length limit, we are not able to cover all aspects. We will consider your suggestions seriously and incorporate them as much as possible.

---

> > ### Public Comment · (anonymous) · 2018-11-28
> > **major concerns remain, REINFORCE still needed**
> >
> > Thank you for beginning to address some of the concerns I raised.
> >
> > However, the major concern about avoiding REINFORCE remains. In particular, computing the gradient of Eq. 13 with respect to the parameters of the generator \theta still requires REINFORCE.
> >
> > Why? Because Eq. 13 involves an expectation over discrete samples s drawn from the generative model G_\theta:
> > \nabla_\theta E_{s ~ G_theta} [ f(s, theta, phi)]
> >
> > How are you actually computing the gradients of Eq. 13 for CoT? My guess is that you are dropping the REINFORCE term in the gradient, and just computing:
> > E_{s ~ G_theta}[\nabla_theta f(s, theta, phi)]
> > which is NOT the gradient of Eq 13.
> >
> > Could you please clarify how you compute this gradient and how you are avoiding REINFORCE?

---

> > > ### Author Response · Authors · 2018-12-05
> > > **Detailed analysis of why REINFORCE is dropped**
> > >
> > > To get better reading experience, you may want to use a LaTeX compiler.
> > >
> > > Eq 13:
> > > \begin{align*}
> > > &\nabla_\theta J_g(\theta)\\
> > > =&\nabla_\theta(\sum_{t=0}^{n-1}\mathop{\mathbb{E}}_{s_t \sim G_\theta} \pi_g(s_t)^T (\log \frac{\pi_m(s_t)}{\pi_g(s_t)})
> > > \end{align*}
> > >
> > > For time step $t$, each term of Eq 13 equals to:
> > > \begin{align*}
> > > &\nabla_\theta J_{g, t}(\theta)\\
> > > =&\nabla_\theta \left[ \mathop{\mathbb{E}}_{s_t \sim G_\theta} \pi_g(s_t)^T(\log \frac{\pi_m(s_t)}{\pi_g(s_t)})\right]\\
> > > =&\nabla_\theta \left[ \sum_{s_t} G_\theta(s_t) (\pi_g(s_t)^T(\log \frac{\pi_m(s_t)}{\pi_g(s_t)})) \right]\\
> > > =&\sum_{s_t} \nabla_\theta \left[ G_\theta(s_t) (\pi_g(s_t)^T(\log \frac{\pi_m(s_t)}{\pi_g(s_t)})) \right]
> > > \end{align*}
> > > Let $L(s_t) = \pi_g(s_t)^T(\log \frac{\pi_m(s_t)}{\pi_g(s_t)})$
> > >
> > > \begin{align*}
> > >   &\nabla_\theta J_{g, t}(\theta) \\
> > >   =&\sum_{s_t}(\frac{\partial G_\theta (s_t)}{\partial \theta} L(s_t) + G_\theta(s_t) \frac{\partial L(s_t)}{\partial \theta}) \\
> > >   =&\sum_{s_t}(G_\theta(s_t)(\frac{\partial \log G_\theta(s_t)}{\partial \theta}L(s_t) + \frac{\partial L(s_t)}{\partial \theta})) \\
> > >   =&\mathop{\mathbb{E}}_{s_t \sim G_\theta} \nabla_\theta (\text{stop\_gradient}(L(s_t))\log G_\theta(s_t) + L(s_t))
> > > \end{align*}
> > >
> > > As you may notice, the total gradient in each step consists of two terms. The first term $(\text{stop\_gradient}(L(s_t))\log G_\theta(s_t)$ behaves like REINFORCE, which introduces variance to the optimization process. The second non-REINFORCE term is comparitively less noisy, though for the first sight it seems not to be working alone.
> > >
> > > If we think about the effects of the two terms, we may notice that they have similar optimization directions (towards minimization of $KL(G_\theta || M_\phi)$ ). Thus, empirically, basic version of CoT introduces an extra hyperparameter $\gamma \in [0, 1]$, to control the balance of the high-variance first term and low-variance second term. The objective in each time step thus becomes:
> > > \begin{align*}
> > >   &\nabla_\theta J^{\gamma}_{g, t}(\theta) \\
> > >   =&\mathop{\mathbb{E}}_{s_t \sim G_\theta} \left[ \nabla_\theta \gamma (\text{stop\_gradient}(L(s_t))\log G_\theta(s_t) + L(s_t)) \right]
> > > \end{align*}
> > >
> > > However, in practice we find that in all our attempts, the algorithm works best when $\gamma = 0.0$. Thus, we directly drop the REINFORCE term. If you think it necessary, we would attach detailed analysis and experiment results in the camera-ready version.

---

> ### Public Comment · (anonymous) · 2018-11-08
> **Addressing:  "(...) and to achieve improved sample quality in terms of BLEU the authors adjust the temperature. Based on the experimental results, it’s not obvious that this set of techniques improves performance over the MLE baseline. "**
>
> I think you are right saying that the evaluation protocol is concerning. Shameless plug: We just wrote a paper on this https://arxiv.org/abs/1811.02549 showing that Textual GANs have wrongly claimed that they can outperform MLE baseline.
> tl;dr : the most effective way to compare NLG models is in quality-diversity space with respect to multiple temperatures. What we find is that MLE outperforms all textual GANs everywhere in the quality-diversity spectrum. MLE is however tied with the newly proposed CoT (see Figure 3 for a comparison of the models on the Oracle task).
>
> That being said, having played with all the language GANs, I can definitely say that CoT works better than all the proposed GANs trained with REINFORCE. However, I am not convinced (yet!) that it outperforms MLE.

---

> > ### Author Response · Authors · 2018-11-10
> > **Response to ''Language GANs Falling Short''**
> >
> > One good property of CoT is that its entropy estimation is more accurate than that of MLE.
> >
> > As is analyzed in some previous work (e.g. Ian's GAN Tutorial in NIPS 2016), models trained to optimize forward KL, which is equivalent to MLE's objective, tend to over-estimate the entropy of the data.
> >
> > In our opinion, the ability of correctly estimating data entropy is important for discrete generative models, since in practice, the entropy of the data is not directly available. For real data, the entropy of the model is not a hyperparameter, but a trainable parameter learned via gradient descent. The manipulation of temperature may help in producing better samples at inference, but it is not a reason to get satisfied in merely doing so.
> >
> > Besides, your synthetic experiment results with MLE and CoT is a bit different from ours, where CoT performs worse and MLE performs remarkably better than that in ours. In our observed results, LMs trained via MLE almost always tends to overfit quickly after about 40 epochs. Without adopting the training techniques you've incorporated in your repo (i.e. Variational Dropout and Many-fold Cross-validation), it is difficult to reproduce your results about MLE, especially for a naive one. While all other models share the same training/testing framework in your code, CoT is absent and instead you used our unfinished repository.
> >
> > Thus, the comparison seems a little bit unfair. While the training techniques for LMs with MLE are well-studied for years, there is much space for investigation of that for LMs trained via CoT (and, of course, discrete GANs). As a result, in our opinion, your experiment shows that CoT with current progress is capable of obtaining comparable results to a well regularized MLE-LM, and even better under a range of entropy settings, where NLL_{oracle} ranges from about 7.1 to 8.8. Please notice that the estimated entropy of the data by CoT given limited observation (10000 samples) lies in such a range. We admit that we are not sure about the reason why CoT cannot keep such advantages in marginal entropy settings (lower and/or higher), but as an educated guess, it may due to that it is more easy for MLE to memorize the training samples while CoT enforces the network to explore-and-improve. As a consequence, if the mediator is not strong enough (i.e. not perfectly matching the assumptions in our theory), the trained model may behave slightly worse in extreme cases.
> >
> > However, despite these minor arguments, in general, we agree with the opinions in your paper and that there needs to be a revolution in the field of language GAN researches before it actually becomes fruitful. Good job!

---

> > > ### Public Comment · (anonymous) · 2018-11-12
> > > **Response to Response**
> > >
> > > We ran your code and created a validation set in order to "early stop" and reported NLL_test on a test set. Because early stopping is a regularization technique, we thus applied some regularization to CoT. We kept the hyperparameters constant thinking that they were the results of a Cross-Validation. If not, would it be possible for you to update the repo with the best performing hyperparameters and then we could rerun the experiment?  Also, does this means that all the MLE results you report are not cross-validated and have no regularization (e.g. dropout)? If so, i'm not sure what kind of conclusions one can come to when outperforming an un-regularized un-cross-validated MLE baseline.
> > >
> > > I'd be happy to continue the conversation via email :)
> > >
> > > FYI, we're coding CoT and adding it to our current repo. We will properly cross-validate CoT on the real dataset EMNLP News 2017 and add CoT in the real data experiment part of our paper.

---

### Author Response · Authors · 2018-11-14
**Paper Revised by the Authors**

The major updates include:
1. Most of the mentioned typos are fixed.
2. The color scheme is changed for better readability in gray scale printing.
3. Standard deviation of the calculated eWMD is provided, in order to make the comparison statistically sound.

---

### Public Comment · (anonymous) · 2018-11-27
**The idea is doubtful, and the proof is erroneous.**

The section explaining the suporiority of training a mediator over training a model P^ to directly predict true distribution P is not convincing. The paper says that there is no guarantee that P^ can be trained to provide correct predictions, but the paper doesn't explain why a mediator doesn't have similar problems.

In the proof of theorem 3 and 4, the paper assumes the mediator M to be trained to be optimal. But it is unjustified. Even if we can train M to be optimal, 2*M-G is already the true distribution P. Why bother to use CoT？

At the same time, I have some suspicions of the experimental results. I think the performance gap is much larger than what CoT can possibly achieve.

---

> ### Author Response · Authors · 2018-11-27
> **Please refer to the submitted version of the paper, instead of the preprint version.**
>
> We've had several updates to address the arguments. Please refer to the submitted version of the paper.

---

### Meta-Review · Area_Chair1 · 2018-12-18
**Novel approach with promising results for generative modeling, but with incorrect claims and insufficiently analysed heuristic shortcuts**

**Confidence:** 4
**Recommendation:** Reject

**Metareview:**

The paper proposes an original and interesting alternative to GANs for optimizing a (proxy to) Jensen-Shannon divergence for discrete sequence data. Experimental results seem promising. Official reviewers were largely positive based on originality and results. However, as it currently stands, the paper still makes false claims that are not well explained or supported, in particular its repeated central claim to provide a "low-variance, bias-free algorithm" to optimize JS.  Given that these central issues were clearly pointed out in a review from a prior submission of this work to another venue (review reposted on the current OpenReview thread on Nov. 6), the AC feels that the authors had had plenty of time to look into them and address them in the paper, as well as occasions to reference and discuss relevant related work pointed in that review. The current version of the paper does neither. The algorithm is not unbiased for at least two reasons pointed out in discussions: a) in practice a parameterized mediator will be unable to match the true P+G, at best yielding a useful biased estimate (not unlike how GAN's parameterized discriminator induces bias). b) One would need to use REINFORCE (or similar) to get an unbiased estimate of the gradient in Eq. 13, a key detail omitted from the paper. From the discussion thread it is possible that authors were initially confused about the fact that this fundamental issue did not disappear with Eq. 13 (they commented "most important idea we want to present in this paper is HOW TO avoid incorporating REINFORCE. Please refer to Eq.13, which is the key to the success of this."). But rather, as guessed by a commentator, that a heuristic implementation, not explained in the paper, dropped the REINFORCE term thus effectively trading variance for bias.
On December 4th authors posted a justification confirming heuristically dropping the REINFORCE terms when taking the gradient of Eq. 13, and said they could attach detailed analysis and experiment results in the camera-ready version.  However if one of the "most important idea" of the paper is how to avoid REINFORCE (as still implied and highlighted in the abstract), the AC finds it worrisome that the paper had no explanation of when and how this was done, and no analysis of the bias induced by (unreportedly) dropping the term.

The approach remains original, interesting, and potentially promising, but as it currently stands, AC and SAC agreed that inexact theoretical over-claiming and insufficient justification and in-depth analysis of key heuristic shortcuts/tradeoffs (however useful) are too important for their fixing to be entrusted to a final camera-ready revision step. A major revision that clearly adresses these issues in depth (both in how the approach is presented and in supporting experiments) will constitute a much more convincing, sound, and impactful research contribution.